



# 3-Dimensional Rockfall Shape Back-Analysis: Methods and Implications

David A. Bonneau[1], D. Jean Hutchinson[1], Paul-Mark Difrancesco[1], Melanie Coombs[1], Zac Sala[1,2]

[1] Department of Geological Sciences and Geological Engineering - Queen's University, Kingston, Ontario, Canada, K7L 3N6
[2] BGC Engineering Inc., Vancouver, British Columbia, Canada, V6Z 0C8

*Correspondence to*: David A. Bonneau (david.bonneau@queensu.ca)

**Abstract.** Rockfall is a complex natural process that can present risks to the effective operation of infrastructure in mountainous terrain. Remote sensing tool and techniques are rapidly becoming the state of practice in the characterization, monitoring and management of these geohazards. The aim of this study is to address the methods and implications of how the dimensions of 3-dimensional rockfall objects, derived from sequential terrestrial laser scans (TLS), are measured. Previous
approaches are reviewed, and two novel algorithms are introduced in an attempt to standardize the process. The approaches are applied to a set of synthetic rockfall objects generated in the open-source software package Blender. In addition, a database of close to 5000 rockfalls is presented derived from sequential TLS monitoring in the White Canyon, British Columbia, Canada. This study illustrates that the method in which the rockfall's dimensions are calculated has a significant impact on how the shape of a rockfall object is classified. This has implications for rockfall modelling as the block shape is known to
influence rockfall runout.

## 1 Introduction

In steep mountainous regions around the world, infrastructure such as highways and railways may be subject to rockfall hazards. A rockfall can be defined as discrete fragments of rock which have detached from a cliff and subsequently fall, bounce and roll as the fragments move downslope by gravity (Hungr et al., 2014). Rockfalls are also characterized by high energy and
mobility, making them a major cause of landslide fatalities (Guzzetti et al., 2004). Moreover, these geohazards can result in economic losses due to service interruptions and equipment damage.

To assist in the management of these geohazards, a rockfall hazard analysis can be undertaken to qualitatively or quantitatively define the rockfall hazard present along a section of linear infrastructure. A typical rockfall hazard analysis involves the
compilation of known rockfall events over a specific spatial scale and within a set period of time (Volkwein et al., 2011). Inventories aim to provide a better understanding about the spatiotemporal occurrence and magnitude of events (Froude and Petley, 2018). Ultimately, temporal trends can be identified from an inventory, which supports a more systematic mapping of





hazards in the region to help mitigate future losses. It may also be useful to discern any long-term changes that are projected, as extreme weather events are expected to increase in both frequency and magnitude within a changing climate (Cloutier et al., 2016).

Once an inventory has been assembled, power law distributions have been suggested to characterize the frequency-magnitude relationship for rockfall at the study slope (Hungr et al., 1999). Using the rockfall frequency-magnitude relationship at the study slope, characterized by specific geological and geomorphological features, return periods for select volume ranges can be determined (Hungr et al., (1999); Dussauge et al. (2003)).

Remote sensing techniques, such as terrestrial laser scanning (TLS), have been used to characterize and monitor rockfall hazards (Abellán et al., 2014; Jaboyedoff et al., 2012; Telling et al., 2017). Change detection algorithms, such as M3C2 (Lague et al., 2013), as an example, can be used to identify areas of loss on slopes (i.e. rockfall) between sequential TLS scans. The location, volume and dimensions of rockfall on the slope can be calculated and populated into a database, as demonstrated by Rosser et al. (2007), Guerin et al. (2014), Tonini and Abellan (2014), van Veen et al. (2017), Janeras et al. (2017) and Williams
et al. (2018). In several of these studies smaller magnitude rockfalls have been identified, which are often not observed during field inspections performed from the base of the slope. Williams et al. (2018) use a fully automated terrestrial laser scanning system to near-continuously monitor a section of coastal cliff in the United Kingdom. With near real-time processing capabilities, they demonstrate the influence of temporal acquisition rate on the calculated frequency-magnitude relationship for rockfall at the study slope. They demonstrate that more frequent monitoring captures a higher proportion of smaller
magnitude rockfall events, which represents a higher frequency magnitude scaling coefficient. However, due to the 2.5D nature of the volumetric analysis, smaller magnitude events resulted in a higher degree of volumetric uncertainty due to edge effects compounded when 3D change maps are converted to 2.5D raster datasets.

Rockfall pre-failure deformation can also be detected and quantified using TLS systems (Abellán et al. 2010, Royán et al.
2014, Kromer et al. 2015a, 2015b). Deformation signatures prior to rockfall events can be associated with entries in rockfall databases to assist with management of rockfall hazards at a given slope (Kromer et al., 2017b; Rowe et al., 2017).

With the rapid automation of TLS acquisition and change detection processing workflows (Kromer et al., 2017a; Williams et al., 2018), we are able to evaluate potential rockfall events and their characteristics quickly and with substantial detail. One of
the key characteristics of the data collected is the rockfall shape and dimensions. A variety of different formulations have been proposed to measure the shape of rockfalls from point cloud datasets, which record the before and after failure geometry. We highlight several methods which have been used in other studies and present two new methodologies to determine the dimensions of a rockfall object.



## 1.1 Rockfall Shape and Dimensions

The quantification of shape of a rockfall scar can provide insight into the kinematics of failure and potential runout of detached material fragments. The use of remote sensing techniques and 3D change detection algorithms permits extraction of true rockfall shape, yet limited work has been completed to quantify shape despite its pronounced effect on runout behaviour
(Glover, 2015; Sala, 2018). Shape, as noted by Blott and Pye (2008), is a function of four primary characteristics which include: form, roughness, irregularity and sphericity. Readers are referred to Blott and Pye (2008) for further details on these characteristics.

In 1958, Sneed and Folk introduced a ternary diagram (Fig. 1A) to describe the shape of pebbles based on relations between
the long (A), intermediate (B) and short orthogonal axes (C). The three ratios are listed below:

$$\frac{C}{A} \tag{1}$$

$$\frac{A-B}{A-C} \tag{2}$$

$$\frac{B}{A} \tag{3}$$

Based on the three relations described above, particles can be classified into ten different shape classes. The end members of the ternary diagram are: compact (cubic), platy (tabular), and elongated (rod shaped).

The Sneed and Folk ternary diagram has been used in rockfall studies to characterize rockfall dimensions (Benjamin, 2018;
van Veen et al., 2017; Williams, 2017). In the aforementioned studies, the method to calculate the dimension of the rockfall is based on a bounding box approach. A bounding box defines the minimum extents of a box which fully encloses the set of points defining the object. In this study, the bounding box is oriented such that the edges of the calculated box are parallel to Cartesian coordinate axes.

Currently there is no standardized method to determine the dimensions of a rockfall object. There is uncertainty in evaluating both the distance and orientation of the axis lengths. This is compounded by the fact that there is ambiguity if the axis measurements are to be mutually orthogonal or not. In this work, we address these uncertainties and propose standardized methods to evaluate the dimensions of a rockfall object.





## 1.2 Objectives

In this work, we address the process of extracting information regarding rockfall dimensions from remotely sensed datasets. The primary objectives of this work are summarized below:

1. Review current approaches used to determine the dimensions of 3D rockfall objects.
2. Present two novel approaches for extracting the dimensions from 3D rockfall objects.
3. Apply all of the approaches to a dataset of synthetic 3D rockfall objects.
4. Implement the proposed approaches on a rockfall database derived from terrestrial laser scanning (TLS) at the White Canyon in the Thompson River Valley in Interior British Columbia, Canada.
5. Determine which method(s) provide the most accurate measurements of the objects' three mutually orthogonal principal axes.

A rockfall object in this context is defined as: a 3-dimensional (3D) point cloud or mesh that approximates the geometry and volume of rock that detached from the slope.

## 1.3 Study Slope – The White Canyon, British Columbia, Canada

The White Canyon (50.266261°, -121.538943°), located in the Thompson Rail Corridor in Interior British Columbia, Canada, is an operationally challenging rock slope (Fig. 2). Rockfall and the movement of debris originating from the steep slopes
present geohazards to the safe operation of the Canadian National (CN) main line, which runs at the base of the slope adjacent to the Thompson River (Bonneau and Hutchinson, 2017; Kromer et al., 2015b; van Veen et al., 2017).

The morphology of the White Canyon is highly complex; differential erosion has formed a morphology which varies across the Canyon and consists of vertical spires and deeply incised channels. The active portion of the Canyon reaches up to 500 m
in height above the railway track. The Canyon spans approximately 2.2 km between Mile 093.1 and 094.6 of the CN Ashcroft subdivision. A series of short portals (tunnels) mark the entrances to the Canyon; a portal can be found on either side of the Canyon. A third portal is located in the middle of the Canyon through a ridge which separates the eastern and western portions of the site.

Two dominant geological units comprise the geology of the White Canyon. The primary unit is the Lytton Gneiss. The Lytton Gneiss is a quartzofeldspathic gneiss with amphibolite bands, containing massive quartzite, amphibolite and gabbroic intrusions (Monger, 1985). In the most western extent of the Canyon, towards the west tunnel portal is the other dominant unit, the Mt. Lytton Batholith. The Mt. Lytton Batholith is a distinctly red stained unit which is composed of granodiorite with local diorite and gabbro. The red staining of the rockmass, is thought to be a direct result of fluids originating from the weathering



of hematite in overlying mid-Cretaceous continental clastic rocks. Two sets of dykes have intruded the Lytton Gneiss, within the White Canyon. The first dyke set consists of tonalitic intrusions which are believed to be related to the emplacement of the Mt. Lytton Batholith (Brown, 1981). The second dyke set is a series of dioritic intrusions which cross cut the Lytton Gneiss and tonalitic dykes. These dioritic intrusions are believed to be part of the Kingsvalle Andesites (Brown, 1981).

## 2 Methodology

As noted above, the methodology involves applying a variety of approaches for measuring geometry of the irregularly shaped blocks to a synthetic dataset generated to represent the range of shapes in the Sneed and Folk ternary diagram (Section 2.1). The methods by which data was collected and processed, using both terrestrial laser scanning (TLS) and Structure-from-Motion Multi-View-Stereo (SfM-MVS) photogrammetry are discussed in Section 2.2. Section 2.3 presents the methodology used to extract rockfall information from 3D change detection. Section 2.4 describes the six approaches used to extract dimension information from 3D point clouds of rockfall.

### 2.1 Synthetic Block Dataset

A synthetic block dataset was generated in the open-source software package Blender (Blender, 2018). Sala (2018) outlines the process used to generate synthetic blocks for rockfall simulation. In general, the process involves the sculpting of 1 m³ cubic meshes. Mesh sculpting in Blender allows for the displacement of mesh geometries into a variety of different shapes, taking into consideration block form characteristics, such as angularity. Once a shape has been created its mesh is subsampled, increasing the number of vertices on the shape's surface to better match the point density which could be achieved from TLS. The mesh vertices are then exported, creating a synthetic rockfall block point cloud. Blocks corresponding to each major class in the Sneed & Folk ternary diagram were created. For each class, (i.e. platy, elongate, cubic, etc.) a rounded and an angular version, as defined by Powers (1953), of the block was generated. Fig. 1B &1C displays examples of the blocks used in this study.

### 2.2 Remote Sensing Data Acquisition

The following subsections (3.2.1 & 3.2.2) outline the remote sensing techniques that are used in this study.

### 2.2.1 Terrestrial Laser Scanning (TLS)

Terrestrial laser scans were taken with an Optech Ilris 3D-ER terrestrial laser scanner (Fig. 1D). The Optech Ilris has a manufacturer-specified accuracy of 7 mm in range from a distance 100 m and 8 mm in vertical and horizontal directions





(Optech, 2014). The maximum range for the Optech Ilris is approximately 800m with 20% target reflectivity (Pesci et al., 2011).

Due to the complex geometrical nature of the White Canyon, several overlapping scans from different vantage points were captured to minimize occlusions and decrease the lateral incidence angle in the scans of the slope. Point spacing for each scan varied between 7 to 10 cm. The scan site locations are displayed in Fig. 3, along with a timeline of the scans used in the study. Scans were taken approximately every 2-3 months starting in November 2014. The last set of TLS scans used in the analysis were taken in December 2017.

To process the TLS scans, the scans were first parsed using Optech Parsing software. Once parsed, vegetation, mesh, and slide detector fences were manually removed from the raw point cloud using PolyWorks PIFEdit. After the point clouds were cleaned, they were aligned using PolyWorks ImAlign to a common baseline (November 2014). The alignment process consisted of a coarse alignment using point picking and then a fine alignment using an iterative closest point (ICP) algorithm (Besl and McKay, 1992). Areas of known change on the slope were excluded from the alignment process to help improve the alignment between sequential scans (Lato et al., 2015).

### 2.2.1 Structure-from-Motion Multi-View-Stereo (SfM-MVS) Photogrammetry

Structure-from-Motion Multi-View-Stereo (SfM-MVS) photogrammetry models were generated of both the Eastern (WCE) and Western (WCW) portions of the White Canyon (Fig. 4). The Agisoft Photoscan Professional V1.3.2 software package (Agisoft LLC, 2018) was used to create the models. The models were generated following a typical SfM-MVS photogrammetry processing workflow (Smith et al., 2016; Westoby et al., 2012).

A Nikon D750 DLSR camera with a Nikkor 50mm f/1.8 prime lens was used for all image acquisitions. An external global position system (GPS) was attached to the camera to geotag each photograph. The images used to generate the White Canyon West (WCW) model were captured on 2018-01-30. The images used to generate the White Canyon East (WCE) model were captured on 2018-04-07. 282 photographs were used to generate the WCW model while 452 photographs were used to generate the WCE model. Images were captured with approximately 50 to 60% overlap.

Each of the SfM-MVS photogrammetry models were remotely mapped to generate masks of the bedrock outcrops and channels in PhotoScan (Fig. 4C). This process is described in detail by (Jolivet et al., 2015). The photogrammetry models and masks were exported and aligned to the TLS datasets in CloudCompare for further analysis. The masks are used in the semi-automated rockfall extraction process that is described below.



## 2.3 Rockfall Extraction Process

In this study, a similar process as utilized by Tonini and Abellan (2014), Carrea et al. (2015), Janeras et al. (2017), van Veen et al. (2017) to semi-automatically identify rockfall locations and extract information related to the dimensions of each rockfall event is implemented. A generalized rockfall extraction process is illustrated in the flow chart in Fig. 5.

The process can be summarized as follows: once the TLS scans are cleaned and aligned, the process involves computing the change between sequential scans taken at times $A$ and $B$. Distances are computed from $A$ to $B$ and then $B$ to $A$. This process determines the front and back of each rockfall event in each respective scan. A minimum change threshold is then applied, this threshold is typically based on the calculated limit of detection. The point clouds of the fronts and backs of all rockfall events

are then merged to generate rockfall objects. Variants of DBSCAN (Ester et al., 1996) are then implemented to cluster individual rockfall events which have occurred between time $A$ and $B$. The dimensions, volume and other parameters of each individual rockfall event can be calculated and then populated into a database for further analysis.

In this study, to compute the change between sequential TLS point clouds, the process outlined in Kromer et al. (2015a) is

utilized. The distance calculation is very similar to M3C2 (Lague et al., 2013), where distances are calculated along normal vectors defined by slope geometry within a specified radius from the point. The change is then filtered based on the limit of detection. The limit of detection (LOD) can be defined based on the registration error (Abellán et al., 2014). In this study, we take two times the standard deviation (95% confidence interval) of the registration error to define the limit of detection. The LOD equates to approximately 5 cm in the summer months and 7 to 10 cm in the winter months (i.e. October to February).

The higher limit of detection in the winter months corresponds to a higher standard deviation in the registration error (alignment). The higher standard deviations corresponds to the winter scans, where there is a higher amount of humidity in the air and possibly water on the slope surface which have all been noted to influence the alignment process (Abellán et al. 2014).

Detectable change was then filtered based on the LOD, to resolve clusters of points that represent the scars (backs) of rockfall

events. This process was repeated, conducting the change detection in the opposite direction to resolve the fronts of the rockfall objects. DBSCAN (Ester et al., 1996) was then used to cluster areas of change. The same parameters as van Veen et al. (2017) are used for the DBSCAN clustering (i.e. search radius of 30 cm and a minimum of 12 points to define a cluster).

To resolve rockfall events as opposed to debris movements, we utilized the masks mapped on the SfM-MVS photogrammetry

models. As opposed to van Veen et al. (2017), who implemented a 2.5D mask, a true 3D mask is used to avoid misclassification. The geometric centroids of each cluster are used to search and find the 10 nearest neighbours within the mask point cloud. Based on the classification of the 10 nearest neighbours within the mask point cloud, a vote is conducted to classify the centroid as either a debris movement or rockfall depending on the mask classification (i.e. bedrock vs. channel).





## 2.4 Model Fitting

The following subsections present the background for each of the models used to determine the dimensions of the rockfall objects. Each of the approaches were implemented in MATLAB (Mathworks, 2018).

### 2.4.1 Bounding Box

A bounding box or enclosing box defines the minimum extents of a box within which all points are contained. In this study, the bounding box is oriented with the edges of the calculated box parallel to the Cartesian coordinate axes (Fig. 6A).

### 2.4.2 Adjusted Bounding Box

The adjusted bounding box approach differs from the bounding box approach in that the orientation of the box is not subjected to any constraints. In this study, singular value decomposition (SVD) (Golub and Loan, 1996) is used to determine the

orientation of the object relative to the principal axes in Cartesian space. SVD is used because this process can handle any $m$ $x$ $n$ matrix whereas eigenvalue decomposition can only be applied to certain classes of square matrices (Golub and Loan, 1996). The direction of most variance using SVD is determined and the point cloud is rotated to align with the direction of maximum variance with the x-axis in Cartesian space. This results in the x-axis of the box being aligned with the longest dimension of the object. A bounding box can then be calculated for the point cloud (Fig. 6B).

### 2.4.3 Least-Squares Ellipsoid

An ellipsoid can be defined as a closed quadric surface that is the analogue of an ellipse. To fit an ellipsoid to the point cloud defining a rockfall object, an algebraic form linear least-squares ellipsoid fit (Schneider and Eberly, 2003) is implemented. An algebraic fitting model was selected as opposed to an orthogonal fitting ellipsoid to reduce computing time and to benefit from the advantages of solving linear least-squares problems (Li and Griffiths, 2004). The algorithm generates a least-squares

ellipsoid fit of the input point cloud (Fig. 6C). Further details on the algorithm and derivation can be found in Schneider and Eberly (2003).

### 2.4.4 Minimum Bounding Sphere

To fit a minimum bounding sphere to the point cloud Welzl's 1991 algorithm is implemented. The algorithm computes the smallest sphere enclosing a set of points in 3-space in linear time (Fig. 6D). For further details on the algorithm, readers are

referred to Welzl (1991).



### 2.4.5 RFSHAPZ

In this study, the RFSHAPZ approach is introduced. The approach can be broken into four main steps: (1) Point cloud preparation, (2) Voxelation, (3) Distance calculations, and (4) Curve fitting. Figure 7 outlines a flowchart for the process used to determine the dimensions of each rockfall object.

Point cloud preparation involves translating each rockfall object so that the object's geometric centroid is centered at the origin of a locally defined Cartesian coordinate system. Once the object is centered at the origin, SVD is used to rotate the object so that the longest dimension is parallel with the x-axis in Cartesian space.

The next step involves generating a voxel grid of the point cloud. A voxel is a 3D volume element that represents a numerical value. For this study, the default voxel cube size is defined as a function of the point spacing. We calculate the average point spacing of the surfaces that make up the rockfall object, and then double the value to determine the voxel size. The size of the voxel is therefore a function of the point spacing and can be adjusted depending on the rockfall object. The voxel grid is used to provide a spatial context for the rockfall object and allows all points within each voxel to be stored for further analysis.

Once the voxel grid is established, for each voxel grid line in the XY and XZ planes, we calculate the maximum Euclidean distance between points within populated voxels (Fig. 6E). The calculated distances are plotted along each grid line. Curves are then fit to each of the distributions, utilizing a Fourier Series fit, a Gaussian fit and a Sum of Sines fit. An overview of each of the fitting methods is provided below.

The Fourier series is a sum of sine and cosine functions that describes a periodic signal. In this study, we use the trigonometric form of the series which can be expressed as:

$$y = a_0 + \sum_{i=1}^{n} a_i \cos(iwx) + b_i \sin(iwx) \tag{4}$$

where $a_0$ is a constant term and is associated with the i = 0 in the cosine term. $w$ represents the fundamental frequency of the signal, and $n$ is the number of terms in the series. For this study, $n$ is fixed at a constant value of one.

The Gaussian model fits peaks in a data series, and is given by:

$$y = \sum_{i=1}^{n} a_i e^{\left[ -\left( \frac{x - b_i}{c_i} \right)^2 \right]} \tag{5}$$





where *a* is the amplitude, *b* is the centroid (location), *c* is related to the peak width, *n* is the number of peaks to fit. For this study, *n* is fixed at a value of one.

The last curve fitting function used is the sum of sines model. This model fits periodic functions, and is given by:

$$y = \sum_{i=1}^{n} a_i \sin(b_i x + c_i) \tag{6}$$

where *a* is the amplitude, *b* is the frequency, and *c* is the phase constant for each sine wave term. n defines the number of terms in the series. This equation is closely related to the Fourier series described in Section 2.4.5.1. The main difference is that the sum of sines equation includes the phase constant and does not include a constant (intercept) term. For this study, *n* is fixed at a value of one.

### 2.4.6 RFCYLIN

The last approach introduced and implemented in this study draws inspiration from the M3C2 methodology (Lague et al., 2013). The point cloud preparation is the same as was described for the RFSHAPZ approach discussed in Section 2.4.5.

For all points in the cloud defining the rockfall object, we calculate the vector and Euclidean distance from each point to the geometric centroid. The vector is oriented towards the calculated centroid. A cylinder is then projected from each point through the geometric centroid of the rockfall object. The length of the cylinder is set to be greater than the distance calculated between each point and the geometric centroid. After the cylinder has been projected, points are found to be within the cylinder. These points are projected on the vector line and the maximum distance between all points through the centroid is determined. This process results in determining the maximum (longest) dimension of the rockfall object.

Once the maximum distance and vector orientation has been calculated, orthogonal vectors to the vector of maximum distance are then calculated through SVD. To do this step, a plane is projected perpendicular to the vector defining the maximum dimension. Points defining the rockfall object are projected onto the plane. SVD is then used to determine the direction of maximum and minimum variance. These define the vector orientations of the other axes. Once the vector orientations of the orthogonal vectors have been determined, cylinders are projected along each vector to find points which lie within the cylinder. If no points are found to be within the cylinder, we incrementally increase the diameter of the cylinder until points are found to be within the cylinder. These points are then projected onto the vector line defining the centerline of the cylinder. The distance between points along each of the orthogonal vectors are calculated and define the intermediate and shortest dimensions of the rockfall object (Fig. 6F). A flowchart outlining this algorithm is displayed in Fig. 8.





## 3 Results

The calculated dimensions of the rockfall objects, using each of the techniques described in Section 2, are tabulated for analysis. Section 3.1 presents the results from the analysis of the synthetic block dataset. Section 3.2 presents the results of the analysis on the rockfall objects extracted from the TLS monitoring in the White Canyon.

### 3.1 Synthetic Block Dataset

The dimensions of the twenty synthetic blocks described in Section 2.1 were measured using the six methods outlined in Section 2.4. Two independent sets of measurements were made manually, to provide a baseline.

The calculated dimensions were plotted on Sneed & Folk ternary diagrams in order to examine the geometric results, as shown
in Fig. 9. The data for the smooth (rounded) and angular synthetic objects is shown on separate diagrams to highlight differences in the distribution of these datasets. Evaluation of the data shows that the calculated measurements for some shapes aligns well, independent of the method applied. The observations made of these data sets include:

- The angular synthetic block dataset displayed the largest spread in the geometry represented by the calculated
dimensions, when compared to the smooth rockfall objects.
- The measured dimensions of the very-bladed and very-elongate blocks, at the bottom left and right corners of the ternary diagram respectively, were closely aligned for all methods and manual measurements.
- The angular compact series (i.e. compact-platy, compact-bladed and compact-elongate), showed the greatest divergence between the manual measurements and the automated methods. A number of the methods, including the
manual measurements, classified the angular compact-platy block as platy. The methods which did correctly categorize this shape, include the bounding box, the adjusted bounding box, the RFSHAPZ – Gaussian fit and the manual measurements. These measurements, however, are not closely aligned, and display significant spread between the data points.
- For the angular compact-elongate block, the two manual measurements incorrectly classify the block as compact
bladed while all of the calculated dimensions classify the block as compact-elongate.

The results of the rounded synthetic block dataset displayed significantly less spread in the calculated and measured block dimensions than the angular counterparts. Only the rounded compact-elongate block had classification issues based on the measured or calculated dimensions. The RFCYLIN approach, RFSHAPZ and adjusted bounding box all classified the block
as compact-bladed.

Taking the manual measurements as the best representation of the dimensions of each of the synthetic blocks, the error in each



dimension measurement from each of the calculated methods (i.e. A, B & C) could be quantified. Figure 10 displays the results for the rounded synthetic blocks and Fig. 11 presents the results for the angular set. The bounding box and adjusted bounding box approaches were excluded from this analysis since they are a component of the process of how the synthetic blocks were generated within Blender (Sala, 2018).

Overall, the errors associated with the angular dataset are an order of magnitude higher than the rounded dataset (A-axis). In addition, for the A-axis measurements, none of the calculated fits underestimated the dimension, for both the angular and rounded datasets. Relative to the rest of the shapes, the platy series (i.e. compact-platy, platy, very-platy), showed the highest deviations from the manual measurement. Within the angular data series, errors on the order of 20 cm were reported for the

A-axis measurement.

## 3.2 White Canyon Rockfall Dataset

Analysis of the TLS data collected at the White Canyon study slope between November 2014 and December 2017, using the semi-automated rockfall extraction process resulted in a database of 4960 rockfall events. 2566 events were identified on the

WCW slopes during the monitoring period, while 2394 events were documented in WCE. The centroid of each of the detected rockfalls is displayed in Fig. 12. The data plotted in this figure displays that the spatial distribution of rockfall is quite varied across the entire canyon. Rockfalls were documented to occur in all lithologies present in the slope.

A sub-set of 50 rockfall events were identified and selected from the overall database for further analysis. As a first pass, only

events larger than 1 m$^3$ were selected from the full database. The resulting 160 rockfall events were considered large enough that a reasonable estimate of their shape could be made from the point cloud where the data points are spaced at approximately 7 cm apart. 110 rockfall events were removed from the sub-set due to their complex, multi-lobed shapes. The remaining 50 events were interpreted to be the result of discrete individual events, based on fairly well constrained shape, relative to rockmass structure present at the rockfall source location. It is probable that numerous smaller failures have occurred from

that same location (van Veen et al., 2017; Williams et al., 2018) during the three to four months elapsed time between scanning campaigns. The change detection, however, will generate the geometry of an apparent single rockfall which is in fact likely to be the result of several coalesced smaller events. Rockfall objects in the database with very complex, multi-lobed morphology were rejected from the sub-set utilized in this analysis. Certainly, we cannot be sure about the number of events that might have contributed to the final rockfall object, unless we have much more frequent scanning intervals. Using the six methods

outlined in Section 2.4, the dimensions of the 50 blocks were measured. In addition to the automated measurements, two sets of independent manual measurements were also made.





Figure 13 displays the Sneed & Folk ternary diagram for each model fit applied to the 50 rockfall cases in the White Canyon. The bounding box approach resulted in a distribution on the Sneed & Folk ternary diagram that is quite scattered, however, the overall trend is towards a more cubic shape for all of the measured rockfall objects. All possible shapes in the Sneed & Folk classification (i.e. compact, very-elongate) are represented by rockfall object shapes assessed using this method.

The results of the other fitting methods and the manual measurements are in stark contrast to the results of the bounding box approach. None of the other fitting methods nor manual measurements classify any of the 50 rockfall events as cubic or in the compact series (i.e. compact-platy, etc.). All the other fitting methods and manual measurements trend towards very-bladed to very-elongate shape classifications, and are distributed across the lower portion of the diagram.

A single rockfall event was isolated from the 50 events to illustrate the complexities inherent in working with real rockfall shapes, as well as the variations in the calculated dimensions (Sneed & Folk shape classification) using each of the methods implemented in the study (Fig. 14). All of the fits used to assess the dimensions of this rockfall event are visually displayed in Fig. 6. The rockfall occurred in the western portion of the White Canyon between June 2015 and August 2015. The rockfall fell from a height approximately 20 m above track level and a number of impact points along the rockfall trajectory were documented from the change detection analysis. The volume of the rockfall event was estimated to be approximately 1.7 m$^3$, and the shape is considered to be well enough constrained that this could be the result of a single event that occurred during that three month period between scans

20 Five different independent manual measurements of the dimensions were conducted for the rockfall object. All of the manual measurements indicated that the rockfall object is being classified as very bladed. The adjusted bounding box, least-squares ellipsoid, RFSHAPZ fits and the RFCYLIN approach all resulted in the rockfall object being classified as very elongate. The bounding box classified the rockfall object as either compact-platy to platy. The spherical fit, as always, classified the rockfall object as compact. This is a direct result of the fact that all calculated dimensions are equal when using the spherical fit.

## 4 Discussion

The shape or form of an object can be classified by assessing aspect ratios of major axes. However, it has been noted that a problem with this approach arises because there is no standardised method used to determine axis length or to assess if the axis measurements should be mutually orthogonal or not. Additionally, there is an inherent ambiguity in selecting the geometric axes of a particle (Blott and Pye, 2008) such that the A axis represents the maximum possible dimension and the B and C axes represent the intermediate and smallest dimensions respectively.



In this study, we have presented and compared six different methods for assessing a rockfall object's dimensions and resulting shape. All of the algorithms have been presented which will permit these approaches to be replicated for future works. In addition, we have created a synthetic dataset of rockfall objects that can be used to assess the effectiveness of new algorithms aimed at determining a rockfall object's dimensions. This represents a step forward in standardizing methodologies for best-

practice in generating remotely sensed rockfall databases.

The results of this study confirm that the method used to measure the rockfall object's dimensions results in significantly different shape classifications. We have demonstrated that a bounding box approach can define a more cubic form of the object, if the orientation of the longest axis of the rockfall object is not parallel with one of the major Cartesian axes. If a bounding

box will be used to determine the object's dimensions, the adjusted bounding box approach should be used instead.

A minimum bounding sphere was shown to be highly inappropriate for dimension extraction in the cases analyzed in this work. The approach results in all dimensions of the object being equal and every single object being classified as compact or equant (i.e. A=B=C). This may be valid for rockfall objects in some narrowly defined geological settings where equant blocks are

released by the slope rockmass. In addition, further work is required to assess the applicability of the minimum bounding sphere approach to assess the dimensions of detached cobbles and boulders from select horizons of postglacial river terraces (Bonneau and Hutchinson, 2018a, 2018b).

The RFCYLIN approach introduced in the study is the most computationally demanding algorithm. The method tries to

standardize an approach to measure dimensions, where each axis is measured orthogonally to one another, after the longest dimension has been defined. However, occlusions and edge-effects in change detection analysis can result in inaccurate distance calculations which affect the object's dimensions. The RFSHAPZ approach attempts to bypass this complication by utilizing curve fitting methods to assess the dimensions even where there is non-uniform point density.

It should be noted that all the automated fits attempt to find the maximum distance defining a dimension. In comparison to the manual measurements, the measured dimension could be reflective of the overall dimension as opposed to the maximum. Increasing angularity or complex features in the shapes can make it increasingly difficult to define orthogonal measurements manually.

The differences in shape classification have direct implications for rockfall modelling. The size and shape of rock blocks affects potential runout trajectories substantially. Shape has been noted to affect the degree to which rolling can be sustained for blocks (Kobayashi et al., 1990). Furthermore, the degree of angularity of a block also has implications for transitions between translational and rotational motion (Pfeiffer and Bowen, 1989).



Industry standard rockfall modelling software packages such as RockyFor3D (Dorren, 2016) still use relatively simple geometric shapes (rectangles, ellipsoids, spheres). Therefore, if the simulation of a cuboid or rectangular prism is considered, where the volume can be defined as a product of the three axes; the measured dimension directly influences the volume of the rockfall being simulated. The volume then defines the mass of the object and as a result, the moment of inertia.

Fityus et al. (2013) found that the size and shape of rockfall debris are statistically different depending on the geological environment. In their study, they measured the three principal orthogonal dimensions of rock block debris with a measuring tape in a variety of geological settings to explore the effect of geology on the size and shape of rockfall debris. They outline that size distribution data used with shape distribution data could be used to undertake stochastic modelling of rockfalls to

estimate the likely trajectories for a rockfall risk assessment. Therefore, depending on how the shape is measured, different shape classes may or may not be used in modelling. In our study, for the 50 rockfall events in the White Canyon, depending on which methodology was used to measure the object's dimensions, significantly different shape classifications were produced. If the bounding box approach is used, all shape classes are represented while all other fits including manual measurements trend towards the very-bladed to very-elongate shapes. Therefore, if forward modelling was to be undertaken,

we might choose to exclude certain shapes depending on the methodology used to calculated rockfall dimensions.

## 5 Conclusions

In this study, we have demonstrated that the method used to measure the dimensions of rockfall objects matters. Depending on the method used, the object's shape may be misclassified into non-representative geometric categories. The classification of shape has implications for rockfall modelling used for the assessment of rockfall hazards. Shape has been shown to have a

large influence on runout distance. Therefore, it is imperative to select a robust method that can accurately and efficiently determine the dimensions of a rockfall object.

As illustrated with the analysis of synthetic blocks, increasing compactness and angularity results in the most difficulties in measuring the dimensions of a rockfall object. All automated methods and manual measurements displayed less scatter for the

rounded dataset in comparison to their angular counterparts. Furthermore, there is a decrease in differences between the calculated dimensions as the object becomes less compact. This is best illustrated with the synthetic very-bladed and very-elongate blocks. The dimensions of both the angular and rounded version resulted in minimal scatter in the calculated dimensions. The differences between the lengths of the long, intermediate and short axes for these blocks are quite apparent. Therefore, both the manual and automated methods can converge on a dimension length and are not subject to the uncertainties

when there are similarities in the length of 2 or 3 of the axes.

The shape of real rockfall objects are quite complex as displayed with the White Canyon dataset. Angularity, non-uniform point spacing and occlusion in the rockfall objects results in complications for exacting dimensions with both manual and automated methods. From the analysis of 50 rockfall events in the White Canyon, it appears that the RFSHAPZ method most closely aligns with the manual measurements. These measurements, however, are still different. Manually measuring the

dimensions of an object can trend towards picking or measuring a representative length as opposed to automated methods which attempt to find the maximum dimensions length. Further to this, we have demonstrated that the methodology used to extract dimensions can vary significantly as demonstrated with the detailed rockfall case from the overall 50 events in the White Canyon.

**Contributions**

David A. Bonneau coordinated the study, generated all algorithms and code, and drafted the manuscript. David A. Bonneau and D. Jean Hutchinson co-analysed the data. Melanie Coombs and Paul-Mark Difrancesco ran code. Zac Sala generated the synthetic block dataset. All co-authors reviewed and edited the manuscript. All authors declare they have no conflict of interest.

**Acknowledgements**

This research was funded by the Natural Sciences and Engineering Research Council of Canada (NSERC) Discovery grants

held by D. Jean Hutchinson and by the Canadian Railway Ground Hazards Research Program (CN Rail, CP Rail, Transport Canada, Geological Survey of Canada). Support was also provided to David A. Bonneau by the NSERC's Graduate Scholarship Program. Past and present Queen's RGHRP team members are greatly acknowledged for their help with data collection.

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



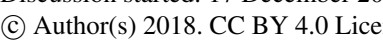



Figure 1. Overview of the Sneed and Folk ternary diagram adapted from Blott and Pye (2008). A) Visual representation of the different shape forms as defined by Sneed and Folk (1958). Inset diagrams display the divisions for each shape class. B) Overview of the rounded synthetic blocks generated using Blender. C) Overview of the angular blocks generated for the study.



**Figure 2. Location map of the White Canyon. A) October 2015 orthophoto of the White Canyon. The White Canyon is delineated by the red dashed line. B) July 2016 panoramic photograph from track level looking northeast at the complex morphology of the study slope. C) July 2016 photograph from track level of the Mt. Lytton Batholith. D) April 2017 photograph displaying the TLS system setup looking at the study slope from across the Thompson River. E) February 2018 photograph from track level looking at one of the rocksheds on the eastern portion of the canyon. The rockshed is 20 m in width.**



**Figure 3. Overview of the scan site locations from across the Thompson River. The timeline across the bottom of the figure indicates the times when TLS scans were captured (green dots).**

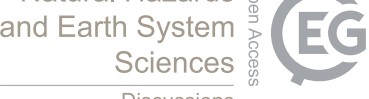



**Figure 4. Overview of the SfM-MVS photogrammetry models. A) Model of WCW taken on January 30th, 2017. B) Model of WCE taken on April 4th, 2017. C) Classified model of WCE. The model was remotely mapped in PhotoScan using a combination of the RGB point cloud and visual inspection of the panoramic photography.**





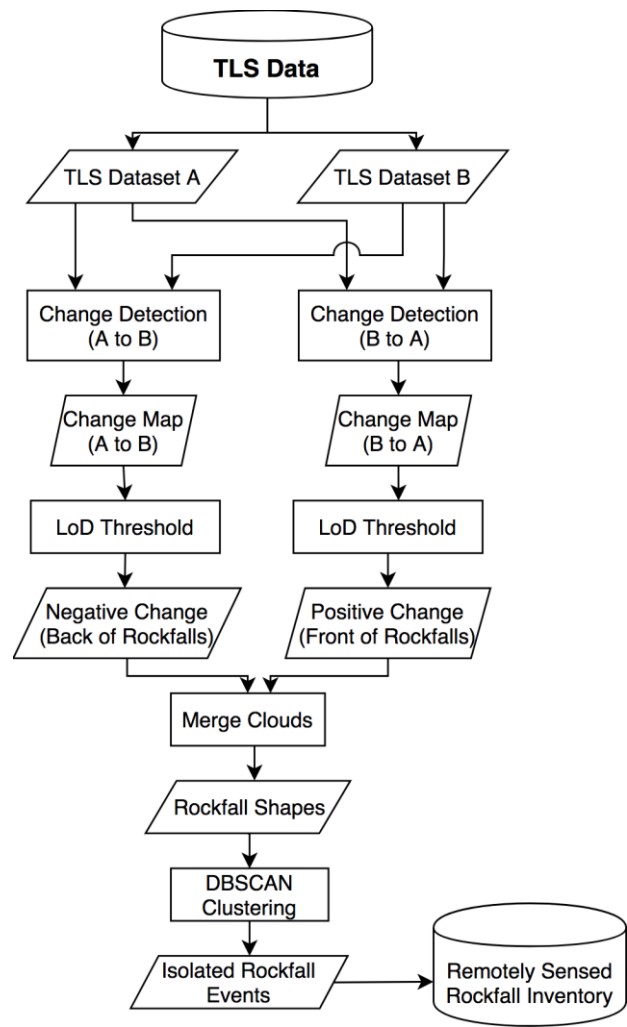

**Figure 5. Structured flow chart of the semi-automated process of extracting rockfall from sequential TLS scans.**

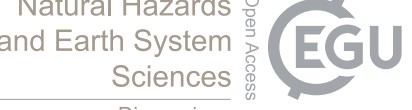

**Figure 6. Visual representation of each of the model fitting methods used in the study. A) Bounding box approach. B) Adjusted bounding box approach. C) Least-squares ellipsoid fit. D) Minimum bounding sphere fit. E) RFSHAPZ approach. F) RFCYLIN approach.**



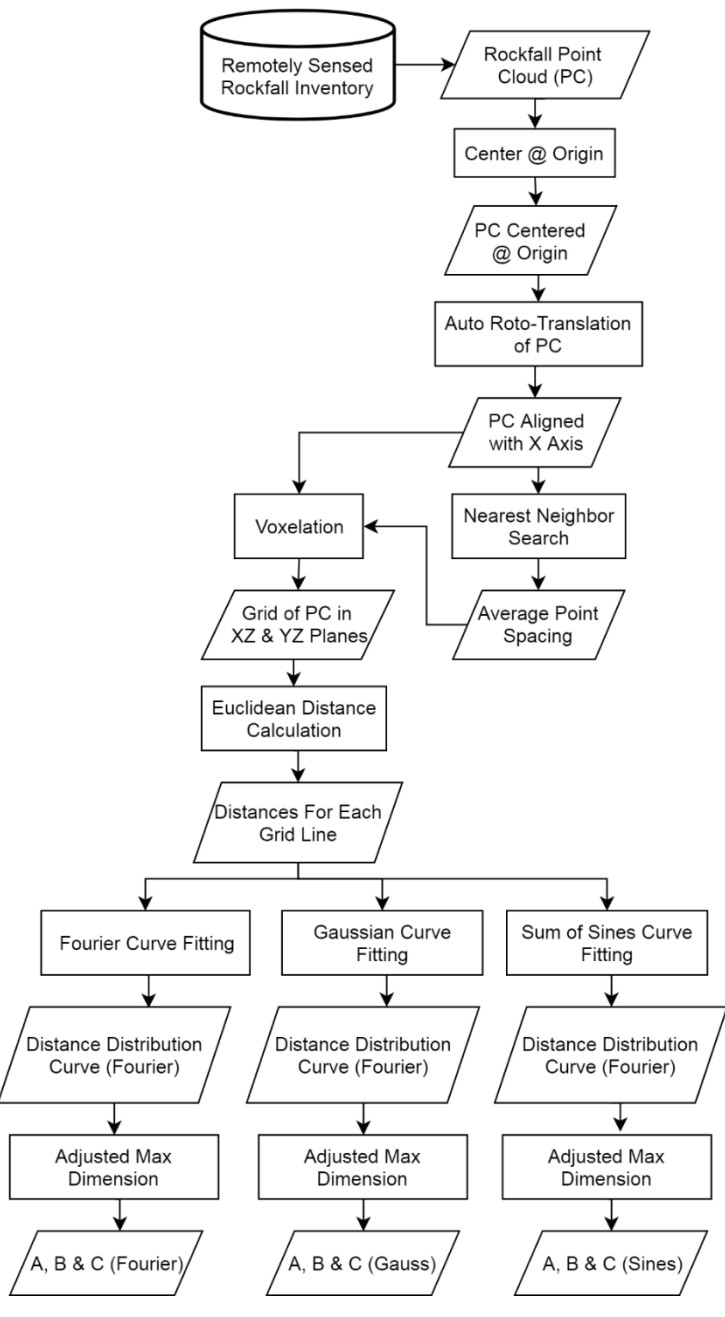

**Figure 7. Structured flow chart of the RFSHAPZ algorithm.**




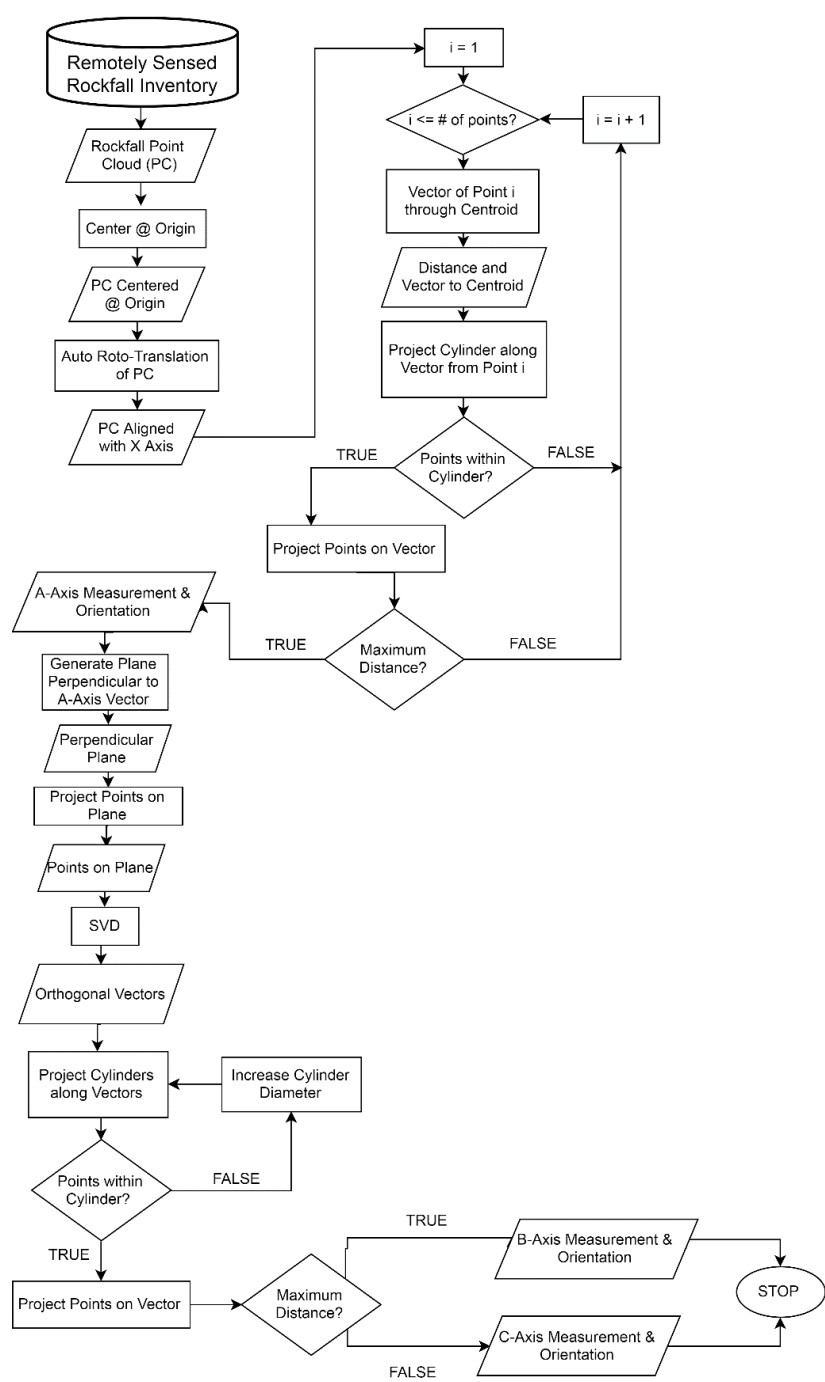

**Figure 8.** **Structured flow chart of the RFCYLIN algorithm.**



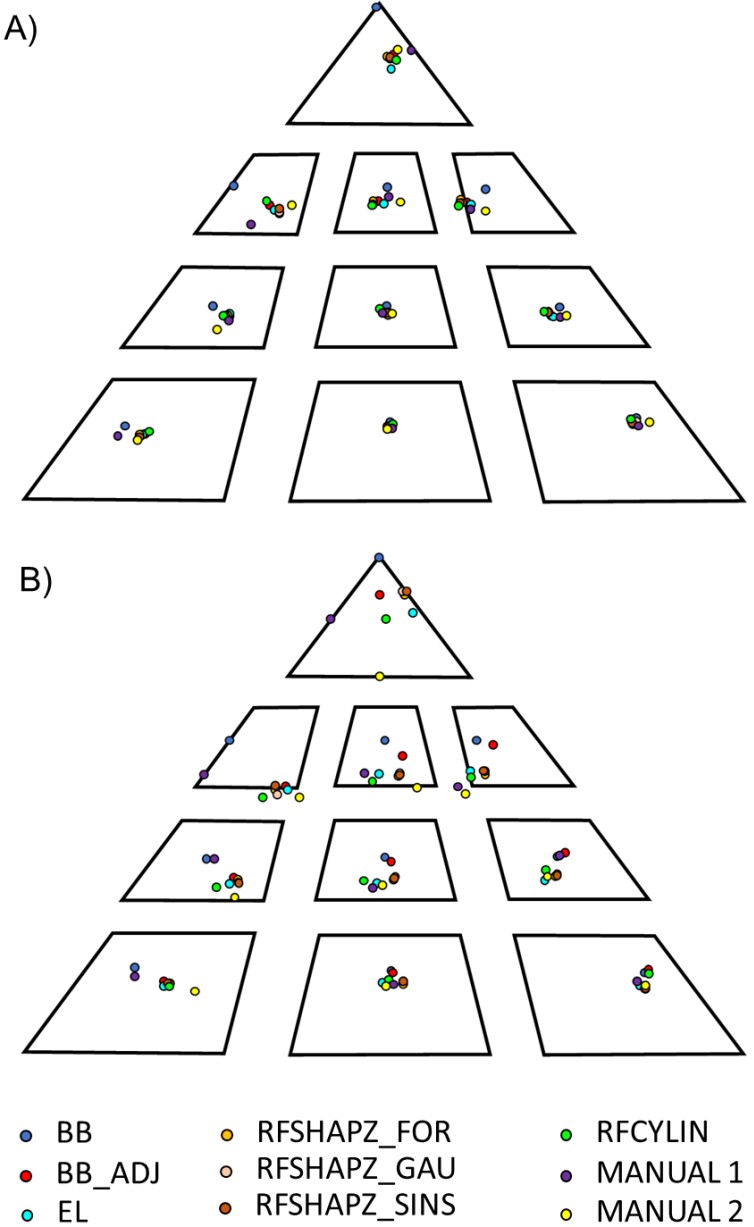

**Figure 9. Sneed and Folk ternary diagrams separated to highlight shape classification results. A) Displaying the results of each of the 9 fits for each of the rounded synthetic blocks. B) Displaying the results of each of the fits for the angular synthetic blocks.**





**Figure 10. Error in dimension measurement for each fit compared to a set of manual measurements for the rounded synthetic blocks.**





**Figure 11. Error in dimension measurement for each fit compared to a set of manual measurements for the angular synthetic blocks.**





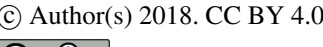

**Figure 12. The White Canyon rockfall database. A) White Canyon West results. The centroid of each rockfall event is displayed as a red dot on the photogrammetry model. B) White Canyon East results.**





**Figure 13. Sneed and Folk ternary diagrams for each of the model fits for the 50 rockfall events that occurred in the White Canyon. Bar chart at the bottom highlights the percentage of classes for each of the fits.**





**Figure 14. Overview of the single rockfall event analyzed. A) Displaying the spatial location of the rockfall event in White Canyon West. The shape of the rockfall object is also displayed. The red points correspond to the front of the object while the blue points correspond to the back of the object. B) the results of the fits for the rockfall object.**

