# Peer review of "3-Dimensional Rockfall Shape Back-Analysis: Methods and Implications"

_Natural Hazards and Earth System Sciences, 2018_

## Referee Comment (RC1) · Anonymous Referee #1 · 26 Feb 2019

The manuscript titled "3-Dimensional Rockfall Shape Back-Analysis: Methods and Implications" by Bonneau et al. reports the results of a research campaign on the methods to define the shape of rockfall blocks. Six approaches are presented in the methodological section and the results are reported on Sneed and Fold triangular plots. The authors demonstrate that the method used to measure the size of the fallen block has certain relevance on blocks' shape, leading to potential misclassification. In parallel, as remarked, rockfall codes largely consider the shape of the block into dynamics calculations and runout distance.

Although the flowchart of the performed tests is clear to me, I have some questions related to the adopted approaches.

1. The authors propose the "Minimum bounding sphere" fitting model (Section 2.2.4). Why the results of the fitting model are not proposed in Figure 13?

2. RFSHAPZ is one of the novel fitting methods. What does "RFSHAPZ" means? Is the method derived from other approaches? Why Fourier series, Gaussian and sum of sines fits? It there a particular reason? Which is the size of the point cloud the authors refer to?

3. RFCYLIN is the other novel fitting method. Similarly, what does "RFCYLIN" means?

4. What differs manual methods 1 and 2?

5. The authors should compare the results of automatized methods to the data obtained through a non-automatized method, say, the manual approach which is considered as "true". This will help in defining the best approach.

---

## Referee Comment (RC2) · Anonymous Referee #2 · 11 Jun 2019

The manuscript is interesting and has important conclusions which could prove useful for authors and studies dealing with 3-D block measurements for rockfall hazard. However - there is some lack of global link and appealing to a larger audience which can help the manuscript become relevant for a larger audience.

The supplement PDF attached contains general and specific comments for the authors.

Please also note the supplement to this comment:
https://www.nat-hazards-earth-syst-sci-discuss.net/nhess-2018-366/nhess-2018-366-RC2-supplement.pdf

2018-366, 2018.

**Supplement:**

Review for paper:
**3-Dimensional Rockfall Shape Back-Analysis: Methods and Implications**
**Author(s): David A. Bonneau et al.**
**MS No.: nhess-2018-366**

**General comments**

The manuscript is interesting and its conclusions may be useful for other authors who deal with 3-D rockfall software simulations for hazard analysis. However – I believe it could be of use to a larger audience if the importance of a very accurate 3-D input dataset to rockfall simulations were better stressed. Perhaps a demonstration of the final simulated rockfall differences between two different output classifications of the suggested methodology on a 3-D based simulation software could emphasize its importance. This could also be achieved by referring to appropriate previous papers.

Additionally, the paper appears to focus on a specific site with little emphasis on its significance for other studies globally. Its localization is also revealed in the missing global reference as to where exactly on the globe is the study area, short acronyms/abbreviations that are not well explained in the text or figure captions, and little comparison to similar results from previous studies (if such exist). Reinforcing the global relevance of the results of this study could benefit to broaden its audience and make it more appropriate for a global publication like NHESS.

**Specific comments**

**Abstract**

It would help the reader to estimate the importance of the paper if the novel approaches or so far un-answered knowledge gaps that the paper suggests to answer are shortly presented in the abstract.

P1 lines 16-17: "a database of close to 5000 rockfalls is presented…" – the authors do not present the database, or any other database which they refer to (50/60 blocks referred to later in the results). Either present the databases in supplemental materials or not state implicitly that you present them.

**Introduction**

**General:** What is the importance of 3-D block mapping for rockfall hazard assessment?

The Introduction could benefit from more reference to previous studies which discuss that issue, and perhaps to point out the importance of the 3-D block classification for rockfall hazard estimation worldwide (as opposed to other simplified block shape methodologies, which were proven to be relatively robust so far). At current stage – this link is missing and the importance of the study is not well constrained.

P1 line 25: perhaps add a short description of rockfall triggers.

P2 lines 5-8: there are more large scale works that can be mentioned as previous works on frequency-magnitude and return periods (e.g. Malamud et al., 2004, Wieczorek and Jäger, 1996).

P2 line 28: what do you mean by 'quickly' and 'with substantial detail'? – how quickly? To what extent of detail?

Section 1.1: rockfall shape were previously described in literature and most rockfall hazard programs use some simple geometry shape for falling blocks – providing relatively good results (e.g. Guzzetti et al., 2002) . Please address what was previously used for hazard assessment and what is the novelty that TLS 3-D measurements (or the current study) contribute to rockfall hazard? Please provide references to emphasize the importance / augmentation of TLS rockfall shape determination over previous methodologies simplifications.

P4 line20: what is the CN main line?

**Results**

In general – there is not a word mentioned on the size of the mapped rockfalls in the results or in the discussion. It will be better to include these – and also power-law size/volume distributions so the readers from other places can relate your database and its applicability to their own study cases and areas.

Section 3.2 – what are the size ranges of the rockfalls in the 160 and the 50 blocks selected?

**Discussion**

In general – the discussion part is relatively short compared to other sections of the manuscript (e.g. Methods or Results). I believe it should be better balanced, as currently the discussion about the insights obtained from the analysis and their implications for other studies were not sufficiently extracted from the data. This could also be achieved by answering the following suggested issues:

Most of the rockfall objects in the study case were classified as 'very bladed' or 'very elongate'. Please consider discussing the possible source for that – i.e. the local geology and structure of the cliff-face or any other factors.

About 30% of the identified 'feasible for analysis' rockfalls were included from the suggested methodology (50 out of 160) due to irregular morphology (not to mention ~4800 excluded cases of less than 1 m^3). Consider discussing the amount of 'good' identified rockfalls valid for using your suggested methodology and relate to how much you assume it is reliable for application in the real-world.

P13 lines 20-24 (Results) + P14 lines 25-27 (Discussion): All suggested automatic methods failed to predict the shape of the single exemplary object (very elongated) with relation to its manual measurements (very bladed). Please consider discussing: (1) the significant contribution or advantage for using the automated methods vs the manual measurements; (2) the significant contribution of the two newly suggested methods in the current study over previously used methods.

Do you consider any scaling factor or effect on your results and conclusions? It appears that most of the discussed rockfalls in the manuscript are of very small size (up to 1-2 cubic meters) compared to other slopes and areas in the world reported in literature (up to tens and even hundreds of cubic meters at places).
Please consider discussing the size of the blocks in the database (volume-frequency power laws) and its implications for larger scale blocks and volumes.

P14 lines 19-24: Consider discussing the superiority of your suggested methodology (if such exists) – how much computation time / effort do these new models require – versus how better is the accuracy they obtain and how significant it is for more successful rockfall hazard estimation? Which one of them would you recommend for use (at least in your case study – and if you can – try to recommend for other readers).

**Conclusion**

Please try to confine the conclusion to insights from the current study only (for example – first paragraph in P15 lines 17-21 cites conclusions from previous studies.)

Please consider actively stating your opinion by suggesting a priority for block shape methodologies: which is most adequate for most cases and which is the less adequate. Try to list them by priority or robustness of success potential to predict real-world rock block shape.

**Figures**
**Figure 2:** Please consider a better World location map for readers outside Canada / N America.

**Figure 6:** Please refer in the figure cation to the relevant studies which presented the different models shown in the plot.

**Figure 9:** As the main results presented in this study – please consider putting more effort in presenting the data more vividly in this plot. There is a lot of white space and very little data presented.

The abbreviations at bottom legend are never referred to in the text or figures. Especially the ones of 'RFSHAPZ_???' should be at least detailed once in the text or figure.

Please add the 'Cubic, Platy, elongated…' the corners of the plots for clarity.

**Figures 10-11:** the abbreviations at right-hand legend are never referred to in the text or figures.

**Figure 12:** please indicate the location of each of the plots (A, B) on Figure 3 of the study area.

What are the sizes or size range of the rockfalls indicated here? It is not mentioned in the text or figures.

How do these sizes relate to the declared identification threshold detailed in the Methods?

**Technical corrections**

The paper is well written in fluent English. I have nothing specific to suggest.

---

## Author Response (AR1)

David A. Bonneau
PhD Student
Queen's University
36 Union St., Kingston, ON, Canada

September 4ᵗʰ, 2019

To: Dr. Andreas Günther
*Editor - NHESS*

I would like to submit the revised manuscript entitled "*3-Dimensional Rockfall Shape Back-Analysis: Methods and Implications*" for your review. Major changes have been implemented to the manuscript based upon the comments of the two reviewers. Notable changes include:

- Condensed the introduction based on suggestions from Reviewer 2.
- Moved the section describing the geology of the White Canyon to Section 3.2. This move was done intentionally to highlight that this study is about the methods used not the results of from the study slope.
- Results was re-written to provide clarity, as suggested by all two reviewers.
- Discussion was improved to discuss the merits of each of the methods.
- Figures 6, 9, 10, 11 & 12 and their respective captions were updated following guidance from reviewers to provide additional clarity of the presented results.
- Additional references suggested by Reviewer 2 were included where appropriate.

All changes made to the manuscript were tracked in the attached word document. The logic for each of the changes are outlined on the following the author's responses to the reviewers. A separate word document includes the manuscript with all the changes accepted.

Correspondence regarding the submission should be directed to the following email and telephone number:

David A. Bonneau
Email: david.bonneau@queensu.ca
Tel: +1 (343) 364-3765

Thank you again for your consideration of our paper.

Sincerely,

David A. Bonneau

**Response to Reviewer 1**

The authors would first like to thank the reviewer for their comments and suggestions.

1. **The authors propose the "Minimum bounding sphere" fitting model (Section 2.2.4). Why the results of the fitting model are not proposed in Figure 13?**

The minimum bounding sphere approach was not included in Figure 13 as it would result in every block plotting at the top of the ternary diagram (see Figure below). In addition, every single block would be classified as cubic. In the minimum bounding sphere approach, all of the calculated dimensions are equal (i.e. A = B = C). However, for completeness, the plot could be included, if required.

[Figure]

Figure 1 – Sneed and Folk diagram presenting the results of the minimum bounding sphere approach for the 50 rockfall objects.

2. **RFSHAPZ is one of the novel fitting methods. What does "RFSHAPZ" means? Is the method derived from other approaches? Why Fourier series, Gaussian and sum of sines fits? It there a particular reason? Which is the size of the point cloud the authors refer to?**

The name RFSHAPZ is used to reflect rockfall (RF) and the derived shapes (SHAPZ). To the authors' current knowledge, the method is not derived from other approaches. Any curve fitting method can be easily implemented in the code which we have developed. For this study, we chose to implement the Fourier, Gaussian and Sum of Sines approaches, to examine the effect of the variation in curve fitting approach each provides. Both the Sum of Sines and Fourier approaches try to fit a curve to a periodic signal. The main difference is that the Sum of Sines equation includes the phase constant and does not include a constant (intercept) term as in the Fourier approach. The Gaussian approach attempts to fit peaks in the data series. The text can be modified to add further clarification.

The authors are not entirely sure what is meant by "…the size of the point cloud the authors refer to?". All of the synthetic blocks were based on sculpting a 1 $m^3$ cubic mesh into the shapes presented in the Sneed and Folk diagram.

3. **RFCYLIN is the other novel fitting method. Similarly, what does "RFCYLIN" means?**

The name is to reflect rockfall (RF) and the use of cylinders (CYLIN) in calculating the dimensions of the rockfall objects. The text will be modified to reflect this description.

4. **What differs manual methods 1 and 2?**

Manual 1 and 2 reflects two different people who manually measured the orientation and dimension of the mutually orthogonal dimensions of the dataset of 50 rockfall blocks from the White Canyon. In Line 30 on Page 12, we state that two sets of independent manual measurements were made. The authors will add further clarification to outline what was measured.

5. **The authors should compare the results of automatized methods to the data obtained through a non-automatized method, say, the manual approach which is considered as "true". This will help in defining the best approach.**

The authors are not sure which cases the reviewer is referring to. In the comparison with the synthetic block dataset, all dimensions calculated with each fitting method were compared to manual measurements of each of the blocks. This comparison formed the basis of the error analysis which was conducted.

In the case of the 50 rockfall blocks from the White Canyon, given the variation in the both sets of manual measurements (Figure 13), the authors were hesitant to define one of the measurements as "true" to compare against. This is further illustrated with the example of the single block in Figure 14. Five different independent manual measurements of the dimensions were conducted for the rockfall object. All of the manual measurements indicated that the rockfall object is being classified as very bladed. The adjusted bounding box, least-squares ellipsoid, RFSHAPZ fits and the RFCYLIN approach all resulted in the rockfall object being classified as very elongate. The bounding box classified the rockfall object as either compact-platy to platy. The spherical fit, as always, classified the rockfall object as compact. This is a direct result of the fact that all calculated dimensions are equal when using the spherical fit. This example hopefully illustrates that automated methods should not be blindly used and the method used should consider the expected block shape given the rockmass structure.

The second reviewer addressed that an addition to the discussion on the methods regarding computation and accuracy should be added. The authors will add and provide recommendations on implementing the different methods.

**Response to Reviewer 2**

The authors would first like to thank the reviewer for their comments and suggestions. The authors would like to point out that there is limited to no work that can be compared to the results of this study. The study was meant to highlight the different methods that can be used to analyze the shape of rockfall events using TLS remote sensing data. This is then demonstrated with application to a case study in the White Canyon, British Columbia, Canada.

In terms of the modelling component, the authors agree with the reviewer. In fact, this is planned future work for another paper. If we must include it here, this will require description of the model, input parameters, parametric analysis and output information, as well as discussions of the study output. This would likely be 6 or more additional pages of text and numerous figures, which will put this well over a reasonable length of paper. We would prefer to keep this work as an additional paper.

**Abstract**
**It would help the reader to estimate the importance of the paper if the novel approaches or so far un-answered**
**knowledge gaps that the paper suggests to answer are shortly presented in the abstract.**

Added in the abstract.

**P1 lines 16-17: "a database of close to 5000 rockfalls is presented…" – the authors do not present the database, or**
**any other database which they refer to (50/60 blocks referred to later in the results). Either present the databases**
**in supplemental materials or not state implicitly that you present them.**

This will be updated in the abstract.

**Introduction**
**General: What is the importance of 3-D block mapping for rockfall hazard assessment?**
**The Introduction could benefit from more reference to previous studies which discuss that issue, and perhaps to**
**point out the importance of the 3-D block classification for rockfall hazard estimation worldwide (as opposed to**
**other simplified block shape methodologies, which were proven to be relatively robust so far). At current stage –**
**this link is missing and the importance of the study is not well constrained.**

**P1 line 25: perhaps add a short description of rockfall triggers.**

Added.

**P2 lines 5-8: there are more large scale works that can be mentioned as previous works on frequency-magnitude**
**and return periods (e.g. Malamud et al., 2004, Wieczorek and Jäger, 1996).**

The suggested publications have been added as references in these lines.

**P2 line 28: what do you mean by 'quickly' and 'with substantial detail'? – how quickly? To what extent of detail?**

What was meant by "quickly" is that a TLS system is quite portable, is mounted on a tripod and can be deployed as soon as the site is accessed. There is no need to establish a baseline data set as is the case with radar systems, for example. In terms of detail, point clouds captured with TLS systems can have sub-millimeter scale point spacings. These sentences will be altered for clarity and references can be added.

**Section 1.1: rockfall shape were previously described in literature and most rockfall hazard programs use some simple geometry shape for falling blocks – providing relatively good results (e.g. Guzzetti et al., 2002). Please address what was previously used for hazard assessment and what is the novelty that TLS 3-D measurements (or the current study) contribute to rockfall hazard? Please provide references to emphasize the importance / augmentation of TLS rockfall shape determination over previous methodologies simplifications.**

STONE (Guzzetti et al., 2002) uses a lumped mass approach to simulate the trajectory of rockfall, such that all of the mass is concentrated in a point. Therefore, the size and shape of the rock being simulated are not considered. More recent work, using rigid body physics (e.g. Sala, 2018) can explicitly capture the interactions of the object's geometry and the terrain. These simulations, including the ones of Glover (2015), show that the true 3D shape can have an influence on the runout of the rockfall object. Utilizing TLS, we can capture the exact shape and location of the block detachment location. These cases can then be explicitly incorporated into rockfall simulations for both calibration of the model and development of more representative hazard mapping.

In comparison to previous approaches for rockfall hazard assessment, TLS offers the ability to capture a large section of slope in great detail. Structural kinematic analysis can be completed using the scan data. The work of Lato et al. (2010) demonstrate a workflow to incorporate TLS into rockfall hazard assessment. The TLS scan provides a permanent record of the slope at a given point in time. With multi-temporal data, change detection can provide insight into locations where rockfall activity is impending or has occurred.

It should also be mentioned that in P3 Lines 2 to 4, the authors reference the works of Glover (2015) and Sala (2018) which demonstrate the effects of rockfall shape on the runout distances when using 3D shapes in numerical simulations.

**P4 line20: what is the CN main line?**

The CN main line is the primary rail track that is situated at the base of the White Canyon. The wording in this sentence has been altered to improve clarity.

**Results**
**In general – there is not a word mentioned on the size of the mapped rockfalls in the results or in the discussion. It will be better to include these – and also power-law size/volume distributions so the readers from other places can relate your database and its applicability to their own study cases and areas.**

Work on the frequency-magnitude relationship has been completed by van Veen et al., (2017) using a subset of the database. The focus of this work was on the shape of the rockfall events, not the frequency-magnitude relationships. The monitoring at the White Canyon study site has not been conducted for long enough time period to generate realistic frequency-magnitude relations for larger volume events, and work is ongoing within the research team on this topic

**Section 3.2 – what are the size ranges of the rockfalls in the 160 and the 50 blocks selected?**

The volumes range from 1 m$^3$ up to 130 m$^3$ as the largest event recorded. This information will be added in the results section.

**Discussion**
**In general – the discussion part is relatively short compared to other sections of the manuscript (e.g. Methods or Results). I believe it should be better balanced, as currently the discussion about the insights obtained from the analysis and their implications for other studies were not sufficiently extracted from the data. This could also be achieved by answering the following suggested issues:**

**Most of the rockfall objects in the study case were classified as 'very bladed' or 'very elongate'. Please consider**
**discussing the possible source for that – i.e. the local geology and structure of the cliff-face or any other factors.**

This is a direct result of the foliation and orientation of the joint sets within the gneiss. As a result, the blocks trend towards the 'very bladed' or 'very elongate' shapes. Details regarding this will be added in the discussion.

**About 30% of the identified 'feasible for analysis' rockfalls were included from the suggested methodology (50out of 160) due to irregular morphology (not to mention ~4800 excluded cases of less than 1 m^3). Consider discussing the amount of 'good' identified rockfalls valid for using your suggested methodology and relate to how much you assume it is reliable for application in the real-world.**

The greater than 1 m$^3$ threshold was set based on a criteria in CN's Rockfall Hazard Rating system (RHRA: Abbott et al., 1998) which focuses on the rockfall events that are greater than 1 m$^3$. This will be clarified in the text.

The reviewer's comment: "amount of 'good' identified rockfalls valid for using your suggested methodology and relate to how much you assume it is reliable for application in the real-world", is not addressed specifically in this paper. This is in part due to the temporal frequency of scanning. It has been shown by a number of authors (e.g. Veen et al. (2017), Williams et al. (2018) and Williams (2017)) that temporal frequency of monitoring has direct implications for the size and shape of the rockfalls that can be detected using remote sensing methods. As longer times elapse between scan intervals, several smaller rockfall events may occur from a location, with the result that the detected shape and volume is larger than would be the case for any of the single events,  The proposed methodologies used in this study will work regardless of the input data and will classify the rockfall shapes accordingly. In more real-world situations, where there is potentially even less data available, large rockfall events could be detected which are in reality a series of smaller coalescing rockfall events. This would then have implications for calculated return period but also the shape. Therefore the concept of "good" is not easily quantified. Ongoing work to characterise rockfall shapes, related to slope geometry and rockfall shapes may yield

some future logic about "good", and to determine which rockfalls, identified from change detection, should be rejected from analysis as they are likely to be the result of coalescence of several events.

The events smaller than 1 m$^3$ were excluded based on the RHRA criteria discussed above. In addition, the authors wanted to use a reasonably manageable subset of the data for a preliminary analysis of the rockfall database in order to demonstrate the methods.

**P13 lines 20-24 (Results) + P14 lines 25-27 (Discussion): All suggested automatic methods failed to predict the shape of the single exemplary object (very elongated) with relation to its manual measurements (very bladed). Please consider discussing: (1) the significant contribution or advantage for using the automated methods vs the manual measurements; (2) the significant contribution of the two newly suggested methods in the current study over previously used methods.**

The exemplary object used for the analysis represents one of the more complicated geometries used in the analysis. This was done purposefully to highlight that in these cases where there is deviation in the shape classification depending on the methods used. The authors have added additional object that is less geometrically complex to demonstrate a case where all the methods align in the shape classification.

**Do you consider any scaling factor or effect on your results and conclusions? It appears that most of the discussed rockfalls in the manuscript are of very small size (up to 1-2 cubic meters) compared to other slopes and areas in the world reported in literature (up to tens and even hundreds of cubic meters at places). Please consider discussing the size of the blocks in the database (volume-frequency power laws) and its implications for larger scale blocks and volumes.**

This database has been collected between a select period of time and we have seen volumes ranging between 1 and 130 m$^3$. We have not been monitoring long enough to see the larger volume events.

There are also a number of recent studies that have demonstrated the influences of temporal frequency of monitoring. van Veen et al. (2017), Williams et al. (2018) and Williams (2017) highlight these considerations where depending on the time-frame analyzed, large rockfall events are actually multiple smaller coalescing events.

**P14 lines 19-24: Consider discussing the superiority of your suggested methodology (if such exists) – how much computation time / effort do these new models require – versus how better is the accuracy they obtain and how significant it is for more successful rockfall hazard estimation? Which one of them would you recommend for use (at least in your case study – and if you can – try to recommend for other readers).**

Considering the definition given by Sneed and Folk (1958) of how to measure shape, the RFCYLIN approach is most closely aligned. The RFCYLIN approach, however, is the most computationally demanding in comparison to the other methods. The RFSHAPZ approach was generated to deal with cases where the surface of the rockfall being analyzed is quite rough and is an attempt at averaging the dimension being calculated.

The bounding box approach should absolutely not be used in any case. All results will be biased towards the cubic end of the Sneed and Folk diagram. If a bounding box type of approach is going to be

implemented, it is necessary to implement the adjusted approach. In addition, the adjusted bounding box approach is one of the simpler methods to implement in comparison to the RFCYLIN and RFSHAPZ approaches.

The authors will elaborate more and provide recommendations on implementing the different methods.

**Conclusion**
**Please try to confine the conclusion to insights from the current study only (for example – first paragraph in P15 lines 17-21 cites conclusions from previous studies.)**

Theses sentences will be altered to improve clarity.

**Please consider actively stating your opinion by suggesting a priority for block shape methodologies: which is most adequate for most cases and which is the less adequate. Try to list them by priority or robustness of success potential to predict real-world rock block shape.**

This has been added.

**Figures**

**Figure 2: Please consider a better World location map for readers outside Canada / N America.**

**Figure 6: Please refer in the figure cation to the relevant studies which presented the different models shown in**
**the plot.**

The appropriate references can be added to the figure caption.

**Figure 9: As the main results presented in this study – please consider putting more effort in presenting the data more vividly in this plot. There is a lot of white space and very little data presented.**

The decision to present the data in this form was deliberate. It was done purposefully to illustrate spatially the difference in using each of the methods to calculate the dimensions and as a result, the classified shape.

**The abbreviations at bottom legend are never referred to in the text or figures. Especially the ones of 'RFSHAPZ_???' should be at least detailed once in the text or figure.**
**Please add the 'Cubic, Platy, elongated…' the corners of the plots for clarity.**

The appropriate abbreviations have been added to the figure caption.

**Figures 10-11: the abbreviations at right-hand legend are never referred to in the text or figures.**

**The descriptions of each of the abbreviations can be included in the figure captions for both Figures 10 and 11.**

Added.

**Figure 12: please indicate the location of each of the plots (A, B) on Figure 3 of the study area. What are the sizes or size range of the rockfalls indicated here? It is not mentioned in the text or figures. How do these sizes relate to the declared identification threshold detailed in the Methods?**

The figure has been updated.

[revised manuscript text omitted]

A)

B)

- BB
- BB_ADJ
- EL
- RFSHAPZ_FOR
- RFSHAPZ_GAU
- RFSHAPZ_SINS
- RFCYLIN
- MANUAL 1
- MANUAL 2

**Figure 9. Sneed and Folk ternary diagrams separated to highlight shape classification results. A) Displaying the results of each of the 9 fits for each of the rounded synthetic blocks. B) Displaying the results of each of the fits for the angular synthetic blocks. BB: bounding box; BB_ADJ: adjusted bounding box; EL: Least-squares ellipsoidal fit; RFSHAPZ_FOR: RFSHAPZ Fourier fit; RFSHAPZ_GAU: RFSHPZ Gaussian fit; RFSHAPZ_SINS: RFSHAPZ Sum of Sines fit; RFCYLIN: RFCYLIN fit.**

[Figure]

**Figure 10. Error in dimension measurement for each fit compared to a set of manual measurements for the rounded synthetic blocks. EL: Least-squares ellipsoidal fit; FOUR: RFSHAPZ Fourier fit; GAUSS: RFSHPZ Gaussian fit; SINES: RFSHAPZ Sum of Sines fit; CYLIN: RFCYLIN fit.**

[Figure]

**Figure 11. Error in dimension measurement for each fit compared to a set of manual measurements for the angular synthetic blocks.**
**SPH: minimum-bounding sphere fit; EL: Least-squares ellipsoidal fit; FOUR: RFSHAPZ Fourier fit; GAUSS: RFSHPZ Gaussian fit; SINES: RFSHAPZ Sum of Sines fit; CYLIN: RFCYLIN fit.**

[Figure]

200 m

150 m

[Figure]

**Figure 12. The White Canyon rockfall database.** The centroid of each rockfall event is displayed as a red dot on the photogrammetry model. The blue dots correspond to the 50 rockfall events analyzed in detail. The light green dots correspond to the events analyzed in Figure 14. **A) White Canyon West results.**  **B) White Canyon East results.**

[Figure]

**Figure 13. Sneed and Folk ternary diagrams for each of the model fits for the 50 rockfall events that occurred in the White Canyon. Bar chart at the bottom highlights the percentage of classes for each of the fits.** BB: bounding box; BB_ADJ: adjusted bounding box; EL: Least-squares ellipsoidal fit; FOUR: RFSHAPZ Fourier fit; GAUSS: RFSHPZ Gaussian fit; SINES: RFSHAPZ Sum of Sines fit; RFCYLIN: RFCYLIN fit.

[Figure]

[Figure]

**Figure 14.** Overview of the single rockfall events analyzed in more detail, with manual measurements made by five different people. A) Displaying the spatial location and shape of the rockfall event in White Canyon West.  The red points correspond to the front of the object while the blue points correspond to the back of the object. B) Displaying the spatial location and shape of the rockfall event in White Canyon East The red points correspond to the front of the object while the blue points correspond to the back of the object. C) Displaying the results of the different fitting methods for the rockfall event shown in (A). D) Displaying the results of the different fitting methods for the rockfall event shown in (B). BB: bounding box; BB_ADJ: adjusted bounding box; EL: Least-squares ellipsoidal fit.